# Is there a "sweet spot" of model complexity for qualitative models used in Ecosystem-Based Management?

Jamie C. Tam[1]*, Sean M. Lucey[2], Alida Bundy[1], Sarah Gaichas[2], Robert Gamble[2]

1 Fisheries and Oceans Canada, Bedford Institute of Oceanography, Dartmouth, Nova Scotia, Canada,
2 Northeast Fisheries Science Center, NOAA, Woods Hole, Massachusetts, United States of America

* jamie.tam@dfo-mpo.gc.ca

## Abstract

Ecosystem models have been developed to help support Ecosystem-Based Management and to help provide better management advice that can account for ecosystem impacts (e.g., climate, species interactions, fishing behaviour). Quantitative end-to-end models have proven to be very useful strategically for exploring future scenarios, but are data intensive, time consuming, and require considerable expertise and training. Conversely, qualitative models have different benefits: they are less dependent on data, relatively faster to develop, can incorporate different types of information that are difficult to measure or combine, and can be co-developed with a variety of audiences. There has been an increase in the use of qualitative models for marine management, however questions have arisen about how well qualitative models perform in comparison to quantitative models, and how they can be used to inform management. Here we compare results from quantitative and qualitative ecosystem models for the same region at differing levels of model complexity to explore their relative utility for EBM. We conclude that the number of linkages between model elements and trophic position of the perturbed model were influential factors in the qualitative model behaviour. When perturbing lower trophic level groups, higher complexity models performed closer to the quantitative model. Lower complexity models were recommended when estimating scenarios with perturbations to mid-trophic groups. Careful consideration among these issues is required to develop the "sweet spot" of model complexity for qualitative ecosystem models to reflect similar results to quantitative models. In addition, utilizing multiple models to determine the strongest impacts from perturbations is recommended to avoid spurious conclusions.

## Introduction

Ecosystem models are quantitative or qualitative representations of social-ecological systems (SES) that can be used to simulate and analyze the dynamics, interactions,

the Creative Commons CC0 public domain dedication.

**Data availability statement:** All relevant data are within the manuscript and its Supporting information files.

**Funding:** The author(s) received no specific funding for this work.

**Competing interests:** The authors have declared that no competing interests exist.

and processes within these systems [1]. They are essential tools to support the spectrum of Ecosystem-Based Management (EBM), including Ecosystem-Based Fisheries Management (EBFM) and Ecosystem Approaches to Fisheries Management (EAFM). Ecosystem models enable scientists and decision-makers to test hypotheses regarding management options that can account for species/functional group interactions, tradeoffs between various stakeholder and management objectives, and adaptation strategies in the face of environmental changes [2–5]. They range from data-heavy, quantitative end-to-end models such as Ecopath with Ecosim and Ecospace or Atlantis, to qualitative conceptual models such as Qualitative Network Models or Bayesian Belief Networks, that are less data-driven representations of ecosystems [1,6–10].

Ecosystem model selection is dependent on the context, objectives, timeline, and capacity related to the EBM question. There are advantages and disadvantages to both quantitative and qualitative models. Quantitative models are useful for capturing full complexity of model parameters and a better understanding of uncertainties within models [1,11,12]. These quantitative models can take a long time to develop (often multiple years) and are reliant on comprehensive datasets with long time series data. In contrast, qualitative models can be generated relatively quickly and have fewer data requirements [13]. Qualitative models can include both environmental or climate parameters (e.g., wind and waves) alongside social-cultural, economic, and governance parameters that are difficult to incorporate into quantitative SES models (e.g., market prices, Local Ecological Knowledge, Traditional Ecological Knowledge). In qualitative modelling, the precision of linkage strengths is ignored and instead they focus on describing the general relationships and trends [13].

Both quantitative and qualitative SES models have been used within EBM approaches to incorporate a wide breadth of ecosystem interactions into fisheries scientific advice. For example, Kaplan et al. [3] used quantitative ecosystem models to explore ecosystem-level Management Strategy Evaluations and Howell et al. [4] used SES information derived from Ecopath with Ecosim models to rescale target fishing mortality from single species models. Qualitative models have been increasingly employed to understand complex marine SES where model elements or linkages are not well understood or lack robust data [6,14–18]. Qualitative models have been used to examine marine SESs under multiple management scenarios [2,19,20].

While fully quantitative models are considered to provide more informative outputs for decision making, questions remain whether qualitative modelling techniques can provide similar information to inform the scientific advice for EBM. Both types of modelling can be useful within decision frameworks (e.g., IEA, Risk Assessment), but comparative information is required to better understand the limitations of qualitative modelling in comparison to quantitative modelling. Olsen et al. [21] compared existing qualitative to quantitative models under different fishing and seal biomass scenarios in four subregions of the North Sea. They determined that correspondence between the qualitative and quantitative models varied between 0–58%, and suggested that the source of variation in correspondence could be linked to model complexity. The study focused on real world examples of quantitative and qualitative models, but not explicitly control for variability or uncertainty between the models.

Here we provide a systematic examination of model complexity while controlling model uncertainty, between quantitative and qualitative models to explore correspondence between models under varying levels of complexity for delivering robust scientific advice to support decision making. Our aim is to better understand how quantitative and qualitative models respond to positive or negative changes, how they may differ and whether such changes vary across model complexity. This will help to determine under what circumstances qualitative models can provide corresponding results to quantitative models and to develop guidance for using qualitative models for EBM.

## Methods

In order to examine the role complexity plays in the correspondence of qualitative and quantitative models we translated an existing quantitative model (Rpath) into a number of qualitative models of varying complexity. We systematically removed linkages within the qualitative models based on the linkage strength from the diet compositions in the Rpath model (Table 1). This reduced the complexity within the models without adding or changing uncertainty between models to compare the outcomes of simulations on simple vs complex ecosystem models. We then ran a series of perturbation experiments through this suite of models and compared the results amongst the quantitative Rpath model from a Bayesian synthesis routine (Ecosense) as well as results from qualitative models using Qualitative Network Modelling (QNM) [15,22,23].

### Quantitative modelling

An existing Ecopath with Ecosim model for the western Scotian shelf and Bay of Fundy, Canada [24] was used as the basis for the analysis. This model was selected as a representative ecosystem within the Northwest Atlantic region, with common ecological attributes between the US and Canada. The model has detailed documentation of its development [25] and has since been used to explore the impacts of climate change [26]. The original model was complex, constructed with 62 functional groups (nodes), 13 of which were modelled as stanzas with 2–4 age groups. Large complex models include more uncertainty, links and potential non-linear responses and can make it more challenging to compare responses across models and scenarios. Therefore, for the purposes of this study, the model was simplified by removing the stanzas (aggregating age classes into one functional group) and, since this analysis was not focused at the species level, single species were aggregated into larger functional groups (e.g., all demersal piscivorous fish were grouped as demersal piscivores). The resulting model had 27 functional groups and one fishery totaling 28 model elements (WSS28; Table SA1 in S1 File.), which is consistent with many other qualitative ecosystem models (e.g., [8,10,13,21]) and with network models in adjacent marine systems [2].

In order to evaluate potential impacts of parameter uncertainty, the WSS28 model was replicated into the R programming software (R Core Team 2019) using the package Rpath [27]. Rpath uses the same base algorithms as Ecopath with Ecosim and includes a Bayesian synthesis routine called Ecosense [22,23,28]. Ecosense uses the base model data

**Table 1. Models used in this study.**

| Model | Linkages | Rationale |
|---|---|---|
| WSS28 Rpath | All | All linkages are captured |
| WSS28 QNM0 | All | All linkages are captured |
| WSS28 QNM10 | Eliminated linkages between −0.10 and +0.10 | Weakest linkages are removed |
| WSS28 QNM20 | Eliminated linkages between −0.20 and +0.20 | Weak linkages are removed |
| WSS28 QNM30 | Eliminated linkages between −0.30 and +0.30 | Weak-moderate linkages are removed |
| WSS28 QNM40 | Eliminated linkages between −0.40 and +0.40 | Moderate-strong linkages are removed |
| WSS28 QNM50 | Eliminated linkages between −0.50 and +0.50 | Only the strongest linkages are captured. |

"pedigree" to generate prior distributions on model inputs, then samples from those model parameter distributions in order to generate alternative parameter sets. Models based on each alternative parameter set are then run through a simulation for 50 years. Parameter sets that allow all species groups to persist 50 years are considered consistent with thermodynamic principles as plausible models and carried forward for further perturbation analysis. This allows users to generate a range of possible outcomes reflecting the uncertainty related to data quality from an otherwise deterministic model. Rpath also facilitated the conversion of the model linkages into the qualitative model as described below (Table 1).

## Qualitative modelling

Qualitative Network Models (QNM) consist of model elements and linkages denoting positive or negative interactions, which are typically abstracted as directed, unweighted, signed digraphs [13]. In the case of the models used in this study, the linkages represent the direction or flow of biomass and are reflective of the energy transfer based on consumption from one group to another. The interaction between model elements may or may not be linear, but there is an assumed overall linear interaction based on the given model structure. A caveat of QNMs is that they require self-limiting loops (negative feedback loops) to stabilize and maintain the system by counteracting deviations from equilibrium [15]. As a result, cannibalism is captured within self-limiting loops, but difficult to distinguish from other parameters that may be self-limiting (e.g., competition). Perturbations can be assessed with QNMs to simulate scenarios on the given system [29]. Model elements of the QNM can be perturbed in a given direction (positively or negatively) implying a small, sustained change. Scenarios were modeled using QPress, a stochastic QNM software package implemented in R [15,30]. QNMs do not allow for numerical prediction; however, they can qualitatively simulate responses within the SES to perturbations [6,15,20].

## Developing models of varying complexity

The quantitative WSS28 Rpath model was translated into the qualitative models of differing complexity using an adjacency matrix to ensure consistency between the two model approaches. This was developed by adding the diet composition matrix and predation mortality (M) matrix of the WSS28 Rpath model with the resulting sum between +1 and −1, where +1 represents 100% of the diet contribution from a prey and −1 represents 100% mortality from a single predator. This developed a single, consolidated value representing both diet and M that is consistent with model elements and connections in the adjacency matrix for the qualitative model. The adjacency matrix was then used as the basis to develop 6 models of different complexity (Table 1). To replicate a practical scenario of qualitative model development, that may not have a full quantitative model as a reference, the complexity was systematically reduced for each qualitative model by removing the linkages between model elements at 10% increments. For example QNM10 retains only linkages above or equal to 10% (Fig 1). With this, each adjacency matrix was then converted into a signed digraph to utilize the functions in QPress. Through this process, the number of linkages were documented going to (inwards) and coming from (outwards) model elements.

## Scenarios

Four perturbation scenarios were examined in all models (Table 2). The scenarios were designed to evaluate food web response to bottom up, middle out, and top down perturbations. All scenarios were tested with both positive and negative perturbations to an influential group at a given trophic level. Bottom up food web scenarios perturbed Phytoplankton at the base of the food web. Middle out scenarios perturbed Small pelagics at an intermediate trophic level supporting managed and protected species. Top down food web scenarios included perturbations to Seals, an apex predator, and Fisheries.

Perturbations for the quantitative model were executed through the Ecosense routine mentioned above. Ecosense resampled model parameters from a uniform distribution around their starting values based on the data pedigree of the

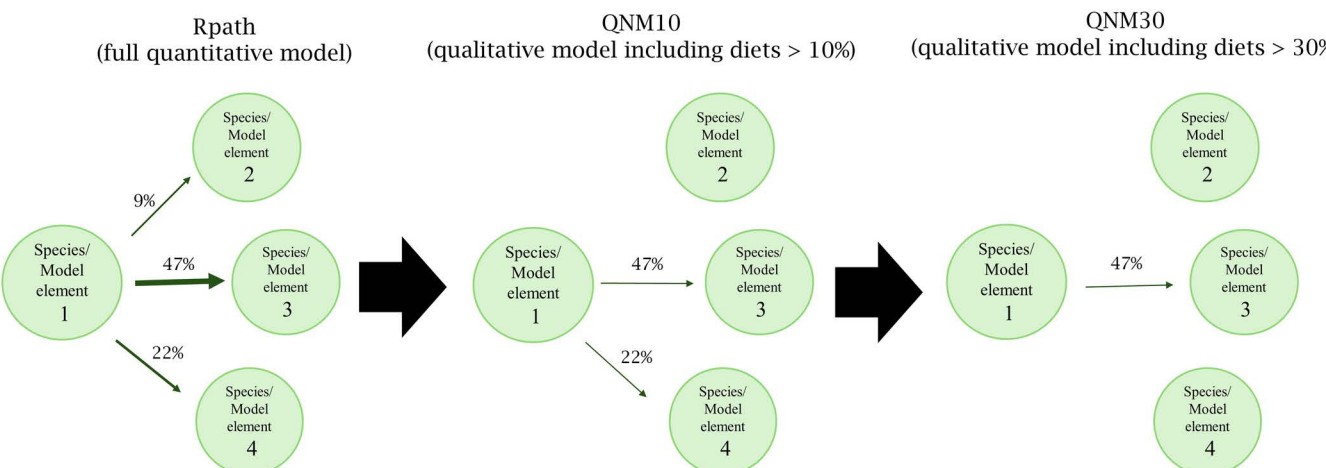

**Fig 1. Conceptual diagram of the development of the qualitative models through the reduction of complexity from the quantitative Rpath model.**

**Table 2. Perturbations used for all models of the study.**

| Model element | Perturbation | Scenarios | |
|---|---|---|---|
| Phytoplankton | Bottom Up | Positive (+) | Negative (-) |
| Small Pelagics | Middle Out | Positive (+) | Negative (-) |
| Seals | Top Down | Positive (+) | Negative (-) |
| Fishing | Top Down | Positive (+) | Negative (-) |

parameter (Table SA2 in S1 File.). In total, 40,000 unique parameter sets were drawn and simulated for 50 years with no perturbations. This resulted in 1153 models that persisted for the entirety of the simulation. The majority of simulations crashed in the first few years and typically failed due to lower trophic level groups going extinct. In order to closely replicate the output from the qualitative models, only the first 1000 plausible parameter sets for both Ecosense and QNMs were used for the perturbation experiments. For all parameter sets a baseline simulation was run in order to measure the effect of the perturbations. The baseline simulation was conducted for 100 years with the average biomass from the last ten used as a reference point for each group. For the Fishery group the average catch of all groups from the last ten years was summed for the reference point.

Perturbations were conducted by using the biomass reference point for the model element to be perturbed (Table 2) and either increasing or decreasing it by 10% using the ForcedBio forcing function in Rpath [27]. This perturbation was applied after an initial ten year ramp up and persisted through the remainder of the simulation (90 years). Perturbations to the Fishery group were done in a similar fashion but used ForcedEffort instead of ForcedBio. The biomass of each group was average over the last ten years of the perturbed simulation and compared to the baseline reference biomass. In order to standardize the magnitude of change between groups with various levels of biomass, a percent difference was calculated as the perturbed biomass minus the baseline biomass divided by the baseline biomass. For the Fishery group summed average catch from the last ten years was used instead of biomass for comparison.

For each group the results from the perturbed run were characterized as either positive or negative. A cutoff of 2% change in either direction was used to determine a positive or negative change. Those values between +2% and −2%

were classified as neutral or no response. Results were then summarized across all perturbed runs as the number of positive, negative, and neutral changes.

Using Qpress (R package) a single perturbation simulation from Table 2 was applied to each of the WSS28 QNM models (Table 1). A matrix is generated from the model by assigning random magnitudes from a predefined distribution (F) to the non-zero interaction coefficients. The stability of the system is then assessed by the eigenvalues of the matrix. If the generated matrix is found to be unstable, the matrix is discarded, and the procedure commences again [15]. Outcomes from the perturbation were recorded for each of the model elements as a marginal likelihood of change (positive, negative, or neutral). The simulations were repeated 1000 times, and marginal likelihoods of changes from the perturbation to each model element were summed for each of the model scenarios.

## Comparisons

For both the qualitative and quantitative approaches, the response of model elements to the perturbation was categorized as positive, negative, neutral, or mixed. These categories were assigned based on the dominant trend of the element increasing, decreasing, or remaining the same among the 1000 simulations, then the number of responses were retained in the results. The categories were based on a majority rule threshold determined by the authors, in a similar way to other qualitative model studies [20]. Responses to the perturbation scenarios could result in any of four outcomes for each model element:

1) Positive response of a model element equal to more than 600 (out of 1000 simulations) positive outcomes on the model element from the scenario. The range of the positive values were recorded (600–1000 simulations).

2) Negative response of a model element equal to more than 600 (out of 1000 simulations) negative outcomes of the model element from the scenario. The range of the negative values were recorded (600–1000 simulations).

3) Neutral responses of an element (no response) equal to more than 600 (out of 1000 simulations) neutral outcomes of the model element from the scenario.

4) Mixed response equal to 600 or less positive and negative (out of 1000 simulations) outcomes of the model element from the scenario indicating a relatively equal positive and negative impacts from a perturbation.

Neutral responses were considered to be the equivalent to no response of a model element in a QPress simulation and within 2% of baseline (start of run) in Rpath. Mixed responses were differentiated from neutral responses due to the level of uncertainty surrounding the response of the model elements to perturbations. While neutral responses indicated a low level of change to a given perturbation, a mixed response showed high uncertainty in the direction of the response to a perturbation.

The level of agreement between model outputs to common perturbations were examined by calculating the percentage of model elements that have a common response to a given perturbation. Comparisons between the qualitative and quantitative model were made after ascertaining the response of the model elements to the perturbation. This was done on a model element-wise basis as well as overall model-to-model performance. In the case of model-to-model comparisons, element-wise comparisons were assigned a point value and summed across all elements (Table 3). Points were assigned based on whether the models agreed with respect to the directionality of their change. For example if both demersal piscivore groups were positive then they were assigned one point. If there was an opposite directionality in change they were assigned negative one point. Half a point was assigned when one of the results was mixed (no dominant direction of change) but leaned towards the same dominant direction as the other model. This occurred when less than 600 but more than 300 perturbation runs were assigned the same direction. Conversely, a model element was assigned a negative half point when the mixed result leaned in the opposite direction of the other model's result. All neutral and truly mixed results did not receive any points.

**Table 3. Point values assigned to model element performance to perturbations from scenarios.**

| Qualitative Result | Quantitative Result | Points Assigned |
|---|---|---|
| Positive | Positive | + 1 |
| Positive | Negative | − 1 |
| Negative | Positive | − 1 |
| Negative | Negative | + 1 |
| Mixed (Positive >300 and <600) | Positive | + 0.5 |
| Mixed (Positive >300 and <600) | Negative | − 0.5 |
| Mixed (Negative >300 and <600) | Positive | − 0.5 |
| Mixed (Negative >300 and <600) | Negative | + 0.5 |
| Mixed (Positive/Negative >300 and <600) | Mixed (Positive/Negative <600) | 0 |
| Neutral | Neutral (<2% from baseline) | 0 |

## Results

### Properties of the quantitative model and QNM models

The full model with 28 model components with full number of interactions (Fig 2A) could be reduced to show only the strongest links (50% linkage strength) without isolating or removing any model elements and retaining all 28 model elements in the ecosystem model (Fig 2B).

There were some trends in the level of influence that the perturbed model elements had across the varying levels of model complexity. While the highest number of linkages occurred for the Rpath and QNM0 models, at QNM30, QNM40, and QNM50, the number of linkages both inwards and outwards of those model elements remained the same for three of the model elements (Phytoplankton, Seals, and Fishery; Table 4). Unsurprisingly, the number and direction of linkages of the perturbed model elements played a role in the results of the perturbations (Fig 3). Phytoplankton, being the lowest trophic functional group, had only outward linkages from other model elements. In contrast, the Fishery had only inward linkages and Seals had predominantly inward linkages with one outward linkage to sharks (Fig 2). Small pelagics, which represented a middle trophic functional group, had a similar number of inward and outward linkages making up the largest total number of linkages among the perturbed elements in the most complex models until QNM40 and QNM50 (Table 4). Linkage strengths greater than 30% were more common at both lowest and highest trophic levels (not shown), resulting in a lower proportion of these linkages lost in the simpler models. Fishery retained the highest proportion of linkages from the full model to the least complex model (to 31%), with the greatest reductions in model linkages observed in Small Pelagics (to 16%).

### Perturbation results

Perturbation results were generally consistent with the known ecology of the system or adjacent systems [8,24,27]. Higher phytoplankton and small pelagics (Phytoplankton+, Small pelagics +) resulted in positive impacts to many model elements across all models of varying complexity, while analogous lower perturbations (Phytoplankton -, Small pelagics -) resulted negative impacts to many model elements (Fig 3). Increased fishing (Fishery +) resulted in negative impacts to some model elements across all models, while decreases to fishery (Fishery -) resulted in higher positive impacts for some model elements (Fig 3). The perturbations to seals (Seals+, -) showed minimal impacts to other model elements, across all the models used in this study with primarily mixed impacts to model elements in QNM0, and neutral impacts to model elements in all other models.

### Rpath vs Qualitative Network Model (QNM0)

The bottom up perturbation of phytoplankton in the Rpath model was transmitted across all functional groups, resulting in an overall increase from the positive perturbation and an overall decrease from the negative perturbation (Fig 3).

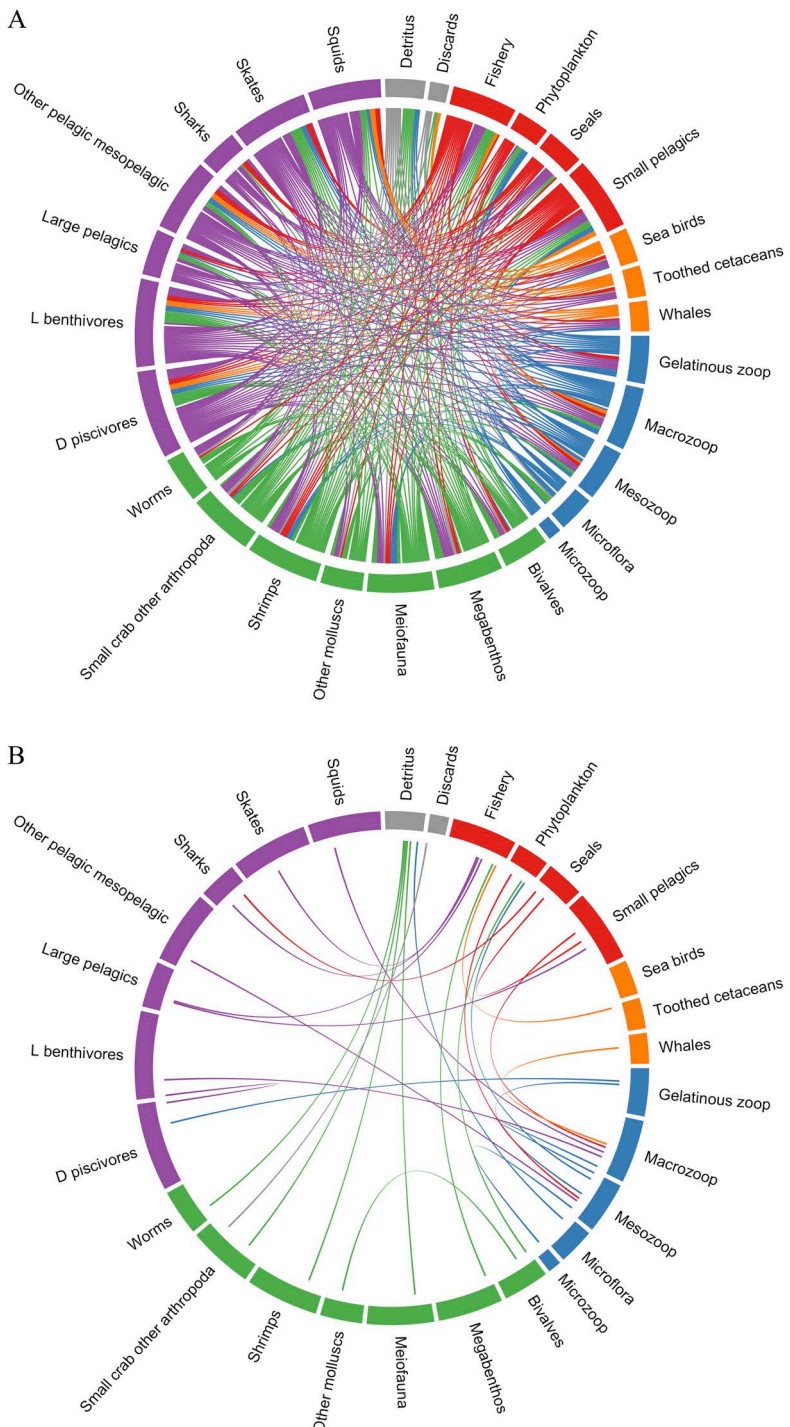

**Fig 2. Circle diagrams showing the linkages between model elements in the A) highest complexity models (Rpath and QNM0) and B) lowest complexity model (QNM50).** While self-limiting loops are in the QNM models, they are not shown in this diagram. Model elements used for the perturbation scenarios: phytoplankton, small pelagics, seals, and fishery are denoted in red.

**Table 4. Number of linkages to each perturbed model element in each of the models examined.**

|  | Linkages | Phytoplankton | | | Small pelagics | | | Seals | | | Fishery | | |
|---|---|---|---|---|---|---|---|---|---|---|---|---|---|
|  |  | IN | OUT | ALL | IN | OUT | ALL | IN | OUT | ALL | IN | OUT | ALL |
| MODEL | Rpath | 0 | 8 | 8 | 7 | 11 | 18 | 8 | 1 | 9 | 16 | 0 | 16 |
|  | QNM0 | 0 | 8 | 8 | 7 | 11 | 18 | 8 | 1 | 9 | 16 | 0 | 16 |
|  | QNM10 | 0 | 4 | 4 | 3 | 8 | 11 | 4 | 1 | 5 | 8 | 0 | 8 |
|  | QNM20 | 0 | 3 | 3 | 2 | 7 | 9 | 2 | 1 | 3 | 7 | 0 | 7 |
|  | QNM30 | 0 | 2 | 2 | 2 | 6 | 8 | 1 | 1 | 2 | 5 | 0 | 5 |
|  | QNM40 | 0 | 2 | 2 | 1 | 4 | 5 | 1 | 1 | 2 | 5 | 0 | 5 |
|  | QNM50 | 0 | 2 | 2 | 1 | 2 | 3 | 1 | 1 | 2 | 5 | 0 | 5 |

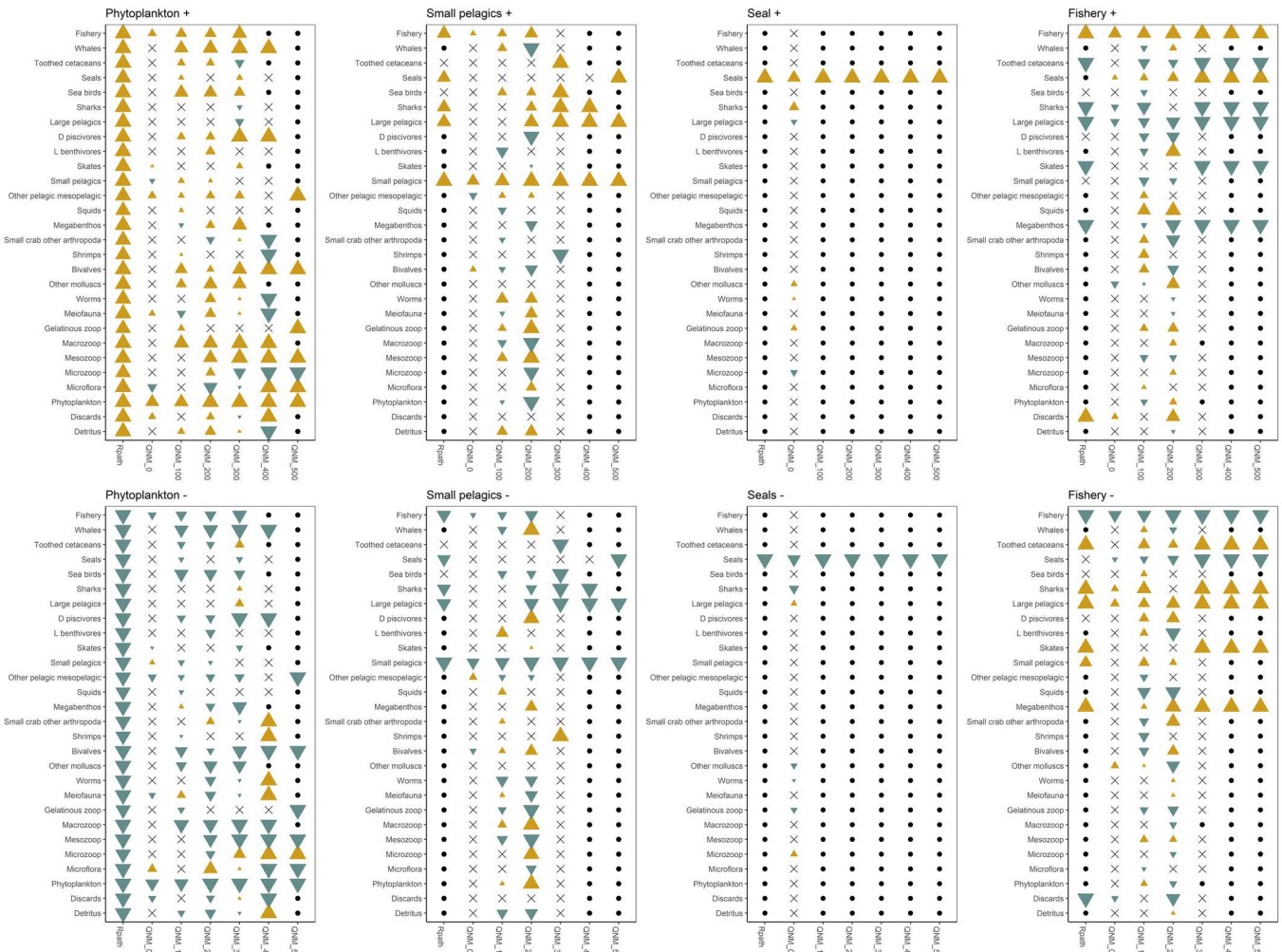

**Fig 3. Model comparisons from each perturbation scenario.** Yellow triangles denote a positive impact from a perturbation scenario on a model element. Green triangles denote a negative impact from a perturbation scenario on a model element. The size of the yellow or green triangles reflect the certainty of the impact of the perturbation on a model element, with larger triangles indicating higher certainty and smaller triangles indicating lower certainty. Black dots denote a neutral impact from a perturbation scenario on a model element. Crosses (x) denote a mixed impact from a perturbation scenario on a model element.

The response for the analogous qualitative model, QNM0, was similar but less pronounced, with positive (or negative) responses observed for the meiofauna, other pelagic mesopelagics, skates, the fishery and discards and negative (or positive) responses for microflora, small pelagics for the positive and negative perturbations respectively.

The middle trophic perturbations were less systemic: large pelagics, sharks, seals and the fishery increased (or decreased) as a result of the positive (or negative) perturbations of small pelagics for Rpath, with neutral responses for most of the rest of the functional groups. The QNM0 results were again less strong and more mixed, with increases (or decreases) observed for bivalves and the fishery and negative (or positive) responses for other pelagic mesopelagics as a result of the positive (or negative) perturbations. Other responses were mixed.

Top down perturbation of seals had little impact in the RPath model; all responses were neutral. The Qpress results were more nuanced, with positive (or negative) responses for sharks, other molluscs, worms and gelatinous zooplankton and decreases (or increases) for large pelagics and microzooplankton as a results of the positive (or negative) perturbations. Other responses were mixed.

Interestingly and unlike the other perturbations, the results for the negative fishery scenario did not completely mirror the positive scenario for the RPath model. In response to the negative perturbation of the fishery, the results for seals and small pelagics were slightly different from those from the positive perturbation of the fishery (Fig 3). These differences were determined to be the consequence of the uncertainty that is accounted for in the Rpath simulations, resulting in the positive and negative outcomes for the seals and small pelagics being binned into adjacent but different categories (neutral and mixed (seals), mixed and positive (small pelagics). The QNM0 results were less strong. An increase (or decrease) in the fishery resulted in a decrease (or increase) in sharks, large pelagics, and other molluscs and an increase (or decrease) in seals and discards. All other responses were mixed.

## Qualitative Network Models (QNM) of varying complexity

With the exception of seals, the complexity of the QNM model impacted the perturbation results. Overall, the impacts of the perturbations on the most complex models (QNM0) were mixed (relatively equal positive and negative impacts to a model element), indicating uncertainty. Models QNM10–30 had the most dynamic responses (elements increasing/decreasing as a result of the perturbations), but as the models were further simplified (QNM 40–50), the impacts of the perturbations became more neutral, indicating more certainty of non-impacts from the perturbation on a model component. This appears to be related to the number and type of linkage (Table 4).

There were nuances in the results however, which varied with the type of perturbation. In the case of phytoplankton, most QNM models showed some consistency in the model elements that increased, decreased or were mixed over the range of complexity, but none of the models produced the consistent bottom-up impacts noted for the Rpath model. The models with most dynamics were QNM20 and QNM30 (mid complexity), which were also the models that were most similar to each other. QNM10 and QNM50 were different from the rest of the models: most of the responses or the former were uncertain and the latter were mostly neutral. The phytoplankton perturbations of the QNM40 and QNM50 models resulted in fewer neutral outcomes than the 3 other types of perturbation (Fig 3), indicating that impacts from phytoplankton were strong (unsurprising for a bottom-up driven model) and propagated through the system.

Positive (or negative) perturbation of the small pelagics resulted in increase (or decrease) of small pelagics in all models and an increase (or decrease) in large pelagics, sharks, seabirds and the fishery in most. QNM10–20 were the most responsive to perturbations, but QNM30–50 were the most similar, with most results neutral. Perturbation of the Fishery resulted in increase (or decrease) in the fishery and seals and a decrease (or increase) in the large pelagics across all models, and a decrease (or increase) in toothed cetaceans, sharks, skates and megabenthos in most models. Similar to the small pelagics, models QNM10–20 were the most responsive to perturbations, but QNM30–50 were the most similar, with most results neutral.

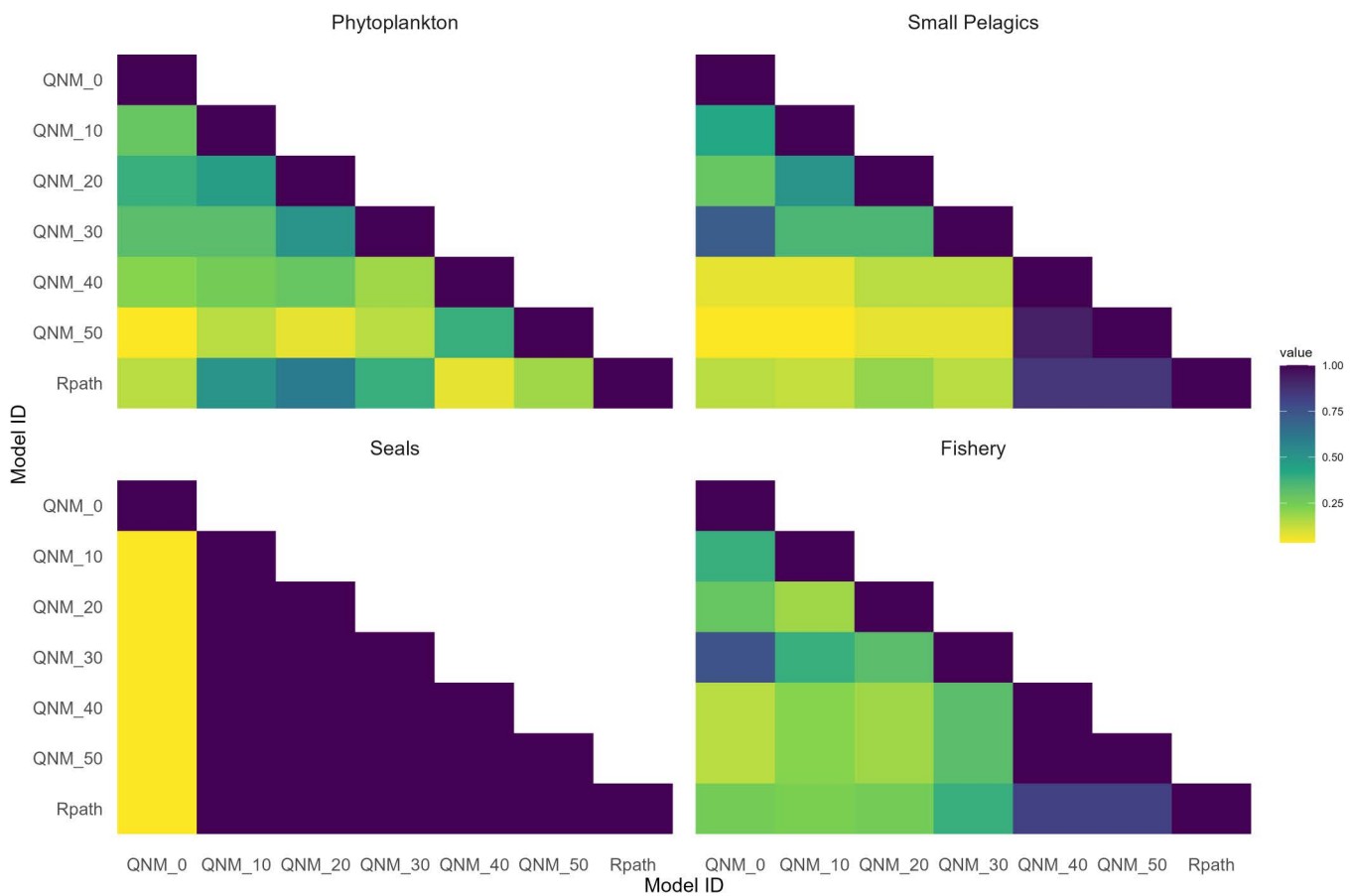

**Fig 4. Agreement plot for all models reflecting the percentage of model elements that have a common response to a given perturbation.** Note that results were the same for positive and negative perturbations, therefore only one plot is shown for each model element.

The perturbation results for seals were different. Seals increased (or decreased) across all levels of model complexity, but the responses of all other model elements, with the exception of the QNM0 model, were neutral: the QNM0 model had mostly mixed responses (Fig 3). This indicates that seals have weaker links and interactions with other model groups than the other perturbed model elements. As a high trophic predator, seals only have outward linkages to sharks and inward linkages decreased to primarily small pelagics as models were simplified. In the WSS ecosystem, small pelagics are preyed on by a number of generalist predators, and it is likely that perturbations to seal have little impact to the rest of the ecosystem, hence mostly neutral impacts as a result of perturbations (Fig 3), and high agreement across all models with the exception of QNM0 (Fig 4).

## Model to model comparison

A comparison of the model-to-model results helped understand where perturbation results across the models were similar, or not (Fig 4). As with the results described above, the degree of agreement among the six models was similar across the positive and negative perturbations for each scenario resulted in similar responses (Fig 4). General agreement among models varied, although as expected, seals provided the highest agreement across models, since these results were mostly neutral or mixed and provided little information regarding the impact of Seals on the rest of the ecosystem. There

was consistently low agreement between Rpath and the most complex QNM model (QNM0) for most scenarios (except seal perturbations) with greater consistency with simpler QNMs (Fig 4). In the case of the Fishery and Small pelagics perturbations, the QNM 40–50 models were most similar to the Rpath model results, and least similar to the other QNM models. However the QNM0–30 models showed a fair degree of similarity, especially in the case of the small pelagics. The perturbation results for Phytoplankton showed a bit of a different pattern. The greatest similarity with the RPath model was observed for the QNM10–30 models, which were also more similar to one another than to the other QNM models, including QNM0 (Fig 4).

## Discussion

As qualitative modelling becomes more prevalent among ecosystem approaches to marine management there has been very little comparative research to evaluate the complexity required of qualitative models compared to quantitative ecosystem models and to evaluate their relative performance. Further, there is little consideration for how model uncertainty can impact qualitative models. This is a necessary area of study as the desire to utilize qualitative modelling for its pragmatic benefits and efficiency is high among resource managers [31]. However to fully apply qualitative modeling to support resource management decisions, there needs to be clearer guidance on how best to develop and use them. Here we explored these questions using a fully quantitative, published trophic model from which we successfully derived six qualitative models of differing complexity, ensuring direct comparability between all models. Unlike other studies, all models represented the same ecosystem for the same time period, using the same data. The intent was to explore real differences in outcomes between quantitative and qualitative models and between models of varying complexity. We had anticipated that we would find a "sweet spot" of model complexity among the six qualitative models, a model of perhaps intermediate complexity that performed best with respect to the quantitative model, across the different scenarios. Instead, we were led to conclude that there needs to be careful consideration regarding the complexity of a qualitative ecosystem model depending on the desired scenarios (perturbations) being examined and the focal model components of interest. While not the "sweeter", straight-forward result that we were expecting, exploring these findings in more detail helped to answer important questions regarding comparative model behaviour (qualitative vs. quantitative).

Only the results for one scenario, Seals, were exactly the same for the quantitative Rpath and qualitative QNM model perturbations; all other perturbations yielded results of varying degrees of difference between all models. Some of these inconsistencies between Rpath and QNM are caused by the fundamental differences in the modelling frameworks. For example, the linkages in the Rpath models are more complex and include weighting of linkage strengths based on diet composition and more nuanced interactions between model components. In addition, parameters for predator-prey functional response can alter Rpath dynamic responses dramatically [32] these parameters were left at default values for this study which tend to stabilize dynamics. Conversely, QNMs were reliant on self-limiting loops that were not necessarily present in Rpath for all groups. Although perturbations through the Ecosense routine in Rpath are most similar to QNMs (as opposed to Ecosim simulations), there were differences in the modelling algorithms and the inherent data uncertainty that account for some of the inconsistency in the impacts to model components [33]. For example Ecosense's Bayesian routine draws from prior distributions based on the data pedigree to account for uncertainty in the generated sets of the quantitative models [22,33]. In contrast QNM models similarly generate sets of the qualitative model, but based on random draws from a normal distribution between zero and one [15].

The number of linkages (connectivity) and trophic position of the perturbed model component in qualitative models is important. When the complexity of the QNM models was highest (QNM0), the agreement with the Rpath model was low and often showed mixed results for many of the model components from perturbations. This aligns with results from Dambacher et al. [13] who concluded that simpler qualitative models can be better representations of the reality of an ecological system because they filter out the noise from interactions that can create feedbacks with counterintuitive effects. This is supported by the results for the most highly linked perturbed model components (> 16 links), Small

pelagics and Fishery. In these cases, the simpler models had high agreement with the Rpath model. For perturbed model components with more moderate linkages (highest complexity 8–9 links, lowest complexity < 2 links), the trophic position highly determined the influence they had when perturbed. Seals, which had only one outward linksshowed very little influence on other model components when perturbed (in Rpath or QNM), and all perturbation results were identical or very similar to the Rpath model. In contrast phytoplankton, which has moderate linkages, all outwards, showed higher influence on other model components and had high agreement between the Rpath model and simpler models (Fig 4). When QNM model components had between 3–5 linkages, the predictability of perturbations was similar to that of Rpath modelling.

When qualitative models are simpler (i.e., fewer linkages between model components), the impacts from a perturbation do not propagate through the ecosystem (fewer linkage pathways). The impact is dampened the further you get away from the perturbed node. This was a consistent observation across all perturbation scenarios for the QNMs. The model elements in simpler qualitative models (QNM40 & QNM50) tended to have consistent neutral responses to perturbations and a higher certainty of stability (no impact to perturbation). The mixed results from phytoplankton and small pelagics perturbations in QNM0 reveal that the impacts of those changes are highly uncertain in these models, likely related to the high number of outward linkages to those perturbed model components).

There were slight differences in the frequency of outcomes between the qualitative and quantitative models. While perturbations to the QNM models resulted in all possible responses (increased, decreased, mixed or neutral), Rpath models show less opposed changes in the ecosystem in response to food-web perturbations. That is, all model components changed either in the direction of the perturbation, neutral or mixed (Phytoplankton, Small pelagics, Seal scenarios), One exception was the negative perturbation in the Fishery scenarios.

This study echoes the recommendations from other applied studies on qualitative ecosystem modelling for resource management in that utilizing multiple approaches or analyses to determine impact to model components is current best practice [8,10,21,34]. While examining the complexity of a qualitative ecosystem model we determined:

1) Generally including linkages based on an estimated diet contribution of 20% yielded the fewest neutral or mixed results. More positive or negative impacts to models resulted from the perturbation at QNM20 for most of the scenarios.

2) Consider including diet components estimated above 10% of the original linkage strength to the qualitative model when examining scenarios with perturbations to lower trophic groups (e.g., phytoplankton).

3) Consider including diet components estimated between 40% − 50% of the original linkage strength to the qualitative model when estimating scenarios with perturbations to mid-trophic groups (e.g., Small pelagics).

4) QNM were not effective at examining impacts of higher trophic model components with few linkages, however similar results were observed for the Ecosense routine.

## Conclusion

These results raise questions about when it is appropriate and useful to use qualitative models to inform resource management. Here we started with qualitative models of different complexity derived from a quantitative model. However, in many cases, research will start with the development of a qualitative model in the absence of a quantitative model, with the focus on key features and relationships of the system or issue of interest. We only explored one type of qualitative model, to keep the analysis simple, but recognise that there are other models that could have been explored, such as Fuzzy Cognitive Maps and Bayesian Belief Networks, with different structures and assumptions that can affect model outcomes [8]. We also recognize that there were other model properties such as examining the impact of latent model linkages or connectance (proportion of linkages) that would be valuable future research in qualitative models. Following best practices, we would recommend a multiple modelling approach, developing qualitative models of different complexity

and, ideally, different types, to embrace model and structural uncertainty. We would also recommend noting that there will likely be no one best model, but a suite of models that can be used together to explore questions of interest.

## Supporting information

**S1 File.** Table SA1. Aggregate groupings from larger WSS63 model to smaller WSS28. Table SA2. Biomass and pedigrees for the WSS28 Rpath model including pedigree for diets (SA3). Table S3. Diet matrix for the Rpath WSS28 model. (DOCX)

## Author contributions

**Conceptualization:** Jamie C. Tam, Sean M. Lucey, Sarah Gaichas, Robert Gamble.

**Formal analysis:** Jamie C. Tam, Sean M. Lucey.

**Methodology:** Jamie C. Tam, Sean M. Lucey, Alida Bundy, Sarah Gaichas.

**Supervision:** Alida Bundy.

**Visualization:** Jamie C. Tam, Sarah Gaichas.

**Writing – original draft:** Jamie C. Tam.

**Writing – review & editing:** Jamie C. Tam, Sean M. Lucey, Alida Bundy, Sarah Gaichas, Robert Gamble.

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
