## [Decision Letter · Decision Letter 0]

PONE-D-25-18645Is there a “sweet spot” of model complexity for qualitative models used in Ecosystem-Based Management?PLOS ONE

Dear Dr. Tam,

Thank you for submitting your manuscript to PLOS ONE. After careful consideration, we feel that it has merit but does not fully meet PLOS ONE’s publication criteria as it currently stands. Therefore, we invite you to submit a revised version of the manuscript that addresses the minor points raised during the review process (note that one of the reviewers made comments directly on the pdf file).

We look forward to receiving your revised manuscript.

Kind regards,

Athanassios C. Tsikliras

Academic Editor

PLOS ONE

Reviewers' comments:

Reviewer's Responses to Questions

**Comments to the Author**

1. Is the manuscript technically sound, and do the data support the conclusions?

Reviewer #1: Yes

Reviewer #2: Yes

2. Has the statistical analysis been performed appropriately and rigorously? 

Reviewer #1: Yes

Reviewer #2: Yes

3. Have the authors made all data underlying the findings in their manuscript fully available?

Reviewer #1: Yes

Reviewer #2: Yes

4. Is the manuscript presented in an intelligible fashion and written in standard English?

Reviewer #1: Yes

Reviewer #2: Yes

5. Review Comments to the Author

Reviewer #1: I have completed my review for the manuscript entitled “Is there a “sweet spot” of model complexity for qualitative models used in Ecosystem-Based Management? (PONE-D-25-18645)” This is a very well-written, conceptually strong, and timely manuscript that contributes meaningfully to the literature on ecosystem-based management (EBM). The authors systematically explore the relationship between model complexity and output reliability in qualitative network models (QNMs) by comparing multiple simplified versions of a well-established quantitative model (Rpath). The core idea, identifying a “sweet spot” of model complexity, is both scientifically interesting and practically useful, especially for decision-makers and stakeholders in resource management contexts. In particular, the study provides an elegant way to balance model interpretability, time/resource efficiency, and robustness, which are often in tension in real-world applications of EBM. This work addresses a real need in applied marine systems modeling, where ecosystem managers often work under constraints that preclude full quantitative modeling. Its relevance aligns with broader ecosystem modelling priorities from ICES and EBM literature that supports multi-model and stakeholder-inclusive approaches. The manuscript is well structured, methods are clearly explained, and the results are robust and ecologically plausible. With a few clarifications and refinements, this work will make a strong contribution.

General comments

The manuscript builds toward identifying an optimal complexity level, but never formally defines what “sweet spot” means (i.e., balance between simplicity and agreement with a quantitative reference). It would be beneficial to include a sentence like: “We define the sweet spot of model complexity as the lowest complexity level that yields agreement with a quantitative model sufficient to support robust scientific inference.”

It would be useful to justify the thresholds for removing linkages in QNM10-50. Providing a rationale grounded in prior work, or at least show a sensitivity check to validate that 10% increments are informative.

Defining a result as positive/negative if ≥600 out of 1000 simulations show consistent direction is a bit arbitrary. I would suggest explaining this as a “majority rule” threshold or provide statistical reasoning (e.g., 95% confidence threshold would be 975+ under binomial assumptions).

The interpretation of mixed vs. neutral needs more rigor. These are currently treated as categorical, yet they reflect different aspects: low effect size vs. high variance. I would suggest considering reporting proportion of runs with positive/negative results for all nodes to clarify this distinction.

Reviewer #2: The article titled “Is there a “sweet spot” of model complexity for qualitative models used in Ecosystem-Based Management?” by Tam et al. compares quantitative and qualitative ecosystem models to discuss the optimal level of model complexity for effective Ecosystem-Based Management advice. The study is an insightful methodological exercise that demonstrates scientific rigor. Thus, I only have few minor comments that are reported in the form of in-text comments in the attached pdf file.

6. PLOS authors have the option to publish the peer review history of their article (what does this mean? ). If published, this will include your full peer review and any attached files.

**Do you want your identity to be public for this peer review?** For information about this choice, including consent withdrawal, please see our Privacy Policy .

Reviewer #1: **Yes: ** Ioannis Keramidas

Reviewer #2: No

---

## [Author Response · Author response to Decision Letter 1]

23 Jun 2025

To Athanassios C. Tsikliras,

Please find our enclosed revised Research Article for PLOS ONE, “Is there a “sweet spot” of model complexity for qualitative models used in Ecosystem-Based Management?”, by Tam et al., for submission [PONE-D-25-18645]. Alongside this response to reviewers, is:

● A marked up copy of the MS that highlights changes to the original version.

● An unmarked version as the revised “Manuscript”.

Thank you for the time and effort to edit and review this paper. The comments have made it much stronger.

We confirm that this manuscript has not been published elsewhere, nor is it under consideration by another journal. All authors have approved the manuscript and agree with the submission to PLOS ONE.

Sincerely,

Jamie C. Tam, PhD

Response to Reviewer #1 (Ioannis Keramidas)

Thank you for your thorough and constructive review. We appreciate your thoughtful assessment of this manuscript and we were happy that you agreed with the need for practical research to support the operationalization of ecosystem models into decision making. Please see the responses to your questions below and have included text clearly labeled in the marked version of the MS.

It would be useful to justify the thresholds for removing linkages in QNM10-50. Providing a rationale grounded in prior work, or at least show a sensitivity check to validate that 10% increments are informative.

Although we agree that a sensitivity might be useful to better understand the specific level to which the linkage strength would impact the overall model, the attempt here was to use a systematic reduction of link strengths rather than to examine thresholds or breakdown points of the model. As we used estimates of diet and mortality from the original Rpath model (which have varying levels of estimation defined in the pedigree table) to determine the linkage strengths, we wanted to consider the practicality of how to use this information to build qualitative models without an existing quantitative ecosystem model. Thus, we attempted to classify major vs moderate vs weak interactions at reasonable increments that would be feasible to estimate interactions under data depauperate scenarios. We determined that 6 qualitative models seemed like a balanced number to examine in this study. Removing linkages above 50% interaction would result in some lost model elements (functional groups) that would no longer be linked. We have made some adjustments in the text at lines 162-166 to clarify these points.

Defining a result as positive/negative if ≥600 out of 1000 simulations show consistent direction is a bit arbitrary. I would suggest explaining this as a “majority rule” threshold or provide statistical reasoning (e.g., 95% confidence threshold would be 975+ under binomial assumptions).

Thank you for this advice. We have included in the text that this was a “majority rule” threshold, as we wanted to depict 50/50 confidently. A difference of 6 runs (e.g. 506) did not seem indicative of a true change, but within 100 (e.g. 600) we felt more confident about the change. This was agreed by consensus by the authors as subject matter experts. We have also included more examples of published work that make similar delineations without using a statistical threshold. Changes were made at lines 227-229.

The interpretation of mixed vs. neutral needs more rigor. These are currently treated as categorical, yet they reflect different aspects: low effect size vs. high variance. I would suggest considering reporting proportion of runs with positive/negative results for all nodes to clarify this distinction.

The results in Figure 3 do show the scale of the results for each model element from each perturbation (size of the points are based on the number of simulation results out of 1000) as we wanted to be transparent about these results. We determined that neutral responses were no change for the node, while mixed nearly equal positive and negative results. We have made these results publicly available in the github that we have now included as part of the supplements.

In the text we have made it clearer that these results are reflecting different aspects of not reacting (neutral) vs reacting in different directions (mixed) to reflect the low effects sizes vs high variances, respectively in lines 326-374.

Response to Reviewer #2

Thank you for your thoughtful review. We appreciate your attention to detail in providing the comments in line. In the text, we have resolved typos, but respond to your in-line comments below:

● Line 131: Link complexity are the same from Rpath to QNM0 Figure 2. Main differences between the base model and the qualitative model is one has link strength, one retains just the direction (positive or negative) as shown in Figure 1. We have adjusted the figure caption to reflect these points.

● Line 153: The values are for individual linkages so representing the relationship between one model element to another model element which represents linkages either as prey and/or predator.

● Line 186-187: We cannot sum the biomass of the fishery as is done for other model elements because fisheries don’t have biomass. We summarized the fishery catch in this way so that it is similarly treated as other functional groups (model elements).

● Line 190: Yes there was no perturbation applied to the first 10 years of the Ecosense runs.

● Line 235: changed to “out of” (typo!)

● Line 318: We realized we were inconsistent here to the previous paragraph where we presented similar results. To remain consistent and avoid confusion, we included “or” in the bracket. This should reflect that the results are mirrored, we discuss positive (or negative). This is to keep the results more concise while remaining accurate.

● Lines 440-448: we removed some of the results details in this section of the discussion to avoid being overly repetitive.

● Line 478: Although fishery is treated here similarly to a functional group, we did not notice much difference in the way that perturbations to the fishery impacted the models compared to other perturbations. We wanted to note here that components with similar linkage structure to seals would not be well examined with the models used in this study.

● Line 491: Yes. We would consider an ensemble model as a possibility, but here considering data depauperate scenarios we thought suites of models might be more appropriate.

● Line 475: Included suggested reference

---

## [Decision Letter · Decision Letter 1]

Is there a “sweet spot” of model complexity for qualitative models used in Ecosystem-Based Management?

PONE-D-25-18645R1

Dear Dr. Tam,

We’re pleased to inform you that your manuscript has been judged scientifically suitable for publication and will be formally accepted for publication once it meets all outstanding technical requirements.

Kind regards,

Athanassios C. Tsikliras

Academic Editor

PLOS ONE

Additional Editor Comments (optional):

Reviewers' comments:

Reviewer's Responses to Questions

**Comments to the Author**

1. If the authors have adequately addressed your comments raised in a previous round of review and you feel that this manuscript is now acceptable for publication, you may indicate that here to bypass the “Comments to the Author” section, enter your conflict of interest statement in the “Confidential to Editor” section, and submit your "Accept" recommendation.

Reviewer #1: All comments have been addressed

Reviewer #2: All comments have been addressed

2. Is the manuscript technically sound, and do the data support the conclusions?

Reviewer #1: Yes

Reviewer #2: Yes

3. Has the statistical analysis been performed appropriately and rigorously? 

Reviewer #1: Yes

Reviewer #2: N/A

4. Have the authors made all data underlying the findings in their manuscript fully available?

Reviewer #1: Yes

Reviewer #2: Yes

5. Is the manuscript presented in an intelligible fashion and written in standard English?

Reviewer #1: Yes

Reviewer #2: Yes

6. Review Comments to the Author

Reviewer #1: I would like to thank the authors for addressing the comments thoroughly and thoughtfully. The clarifications provided on the rationale for linkage reduction thresholds and the use of the “majority rule” criterion strengthen the methodological transparency of the manuscript. I find the revisions satisfactory and believe the manuscript is now suitable for publication.

Reviewer #2: I thank the authors for considering my minor comments.

7. PLOS authors have the option to publish the peer review history of their article (what does this mean? ). If published, this will include your full peer review and any attached files.

**Do you want your identity to be public for this peer review?** For information about this choice, including consent withdrawal, please see our Privacy Policy .

Reviewer #1: **Yes: ** Ioannis Keramidas

Reviewer #2: No

---

## [Editor Report · Acceptance letter]

PONE-D-25-18645R1

PLOS ONE

Dear Dr. Tam,

I'm pleased to inform you that your manuscript has been deemed suitable for publication in PLOS ONE. Congratulations! Your manuscript is now being handed over to our production team.

Kind regards,

on behalf of

Professor Athanassios C. Tsikliras

Academic Editor

PLOS ONE